# Latent Tuberculosis Infection among Healthcare Workers in Duhok Province: From Screening to Prophylactic Treatment

**DOI:** 10.3390/tropicalmed4020085

**Published:** 2019-05-23

**Authors:** Hind Bahzad Almufty, Ibtesam Salih Abdulrahman, Muayad Aghali Merza

**Affiliations:** Department of Clinical Pharmacy, College of Pharmacy, University of Duhok, Duhok 42001, Iraq; bahzadhind@yahoo.com (H.B.A.); ibtesam.abdulrhman@uod.ac (I.S.A.)

**Keywords:** healthcare workers, latent tuberculosis infection, Duhok province

## Abstract

Healthcare workers (HCWs) are at increased risk of infection with *Mycobacterium tuberculosis* (*Mtb*) and, hence, of developing tuberculosis (TB) disease. The aims of this study are to identify the prevalence and determinants of latent TB infection (LTBI) among HCWs in Duhok Province. This is a cross-sectional prospective study conducted during April–July 2018 in different health care facilities of Duhok province. HCWs at multiple levels were selected by a non-systematic random sampling method. Information on demographic and associated risk factors of LTBI were collected by using a standardized questionnaire. Thereafter, all HCWs underwent QuantiFERON Gold Plus (QFT-Plus) assay. HCWs with indeterminate QFT-Plus underwent a Tuberculin Skin Test. HCWs with positive results were further evaluated by smear microscopy investigation and chest X-ray examination. Three hundred ninety-five HCWs were enrolled; 49 (12%) tested positive for LTBI. The mean age of the HCWs was 33.4 ± 9.25 with a female predominance (51.1%). According to the univariate analysis, LTBI was significantly higher among HCWs with the following: age groups ≥ 30 years, alcohol intake, ≥ 11 years of employment, high risk stratification workplaces, and medical doctors. In the multivariate analysis, the age group of 30–39 years (OR = 0.288, 95% CI: 0.105–0.794, *p* value = 0.016) was the only risk factor associated with LTBI. Further medical investigations did not reveal active TB cases among HCWs with LTBI. With regards to prophylactic treatment, 31 (63.3%) LTBI HCWs accepted the treatment, whereas 18 (36.7%) declined the chemoprophylaxis. Of these 31 HCWs on chemoprophylaxis, 12 (38.7%) received isoniazid (INH) for six months, 17 (54.8%) received INH in combination with rifampicin (RMP) for three months, and two (6.5%) received alternative therapy because of anti-TB drug intolerance. In conclusions, although Iraq is a relatively high TB burden country, the prevalence of LTBI among Duhok HCWs is relatively low. It is important to screen HCWs in Duhok for LTBI, particularly medical doctors, young adults, alcoholics, and those whom had a long duration of employment in high-risk workplaces. The acceptance rate of HCWs with LTBI to chemoprophylaxis was low. Therefore, ensuring medical efforts to educate the healthcare staff particularly, non-professionals are a priority to encourage chemoprophylaxis acceptance.

## 1. Background

Tuberculosis was declared a global health emergency for more than two decades by the World Health Organization (WHO) [1]. Approximately, 25% of the global population is infected with latent TB infection (LTBI). The LTBI is described as “a state of persistent immune response to stimulation by *Mycobacterium tuberculosis* (*Mtb*) antigens without evidence of clinically manifested active TB” [2]. A small proportion of LTBI, around 10%, especially within the first few years may progress to active TB disease [3]. There are certain risk factors that could potentiate the reactivation of LTBI, namely patients with HIV coinfection, hemodialysis, immunosuppressive therapy, organ transplantation, malignancy, diabetes, alcoholism, cigarette smoking, underweight, and malnutrition [3,4,5]. Healthcare workers (HCWs) are other victims with an increased risk of LTBI and TB disease, because of their sustained occupational exposure to such infection [6]. The situation is more perplexed in healthcare facilities with poor infection control measures. Local studies from Iraqi Kurdistan showed a very low prevalence of HIV infection among both the general population and specific populations, such as HCWs [7,8]. 

The diagnosis of LTBI can be made by measuring immune response to *Mtb* antigen as detecting the bacilli is not possible. Currently, there are two tests that can identify LTBI. The first one is the conventional Tuberculin Skin Test (TST), which has been used for more than a century with specificity limitations as it may cross-react with Bacillus of Calmette and Guerin (BCG) and Mycobacterium Other Than TB (MOTT) infection. The second test, called Interferon Gamma Release Assay (IGRA), is a well-developed laboratory technique in measuring interferon gamma from T cells following stimulation by *Mtb* complex. As a result, IGRA overcame the limitations of TST [9]. 

In countries with high TB prevalence, the WHO Expanded Program on Immunization recommends BCG vaccination during the first week of birth. In Iraq, as a part of national immunization program, BCG vaccine is routinely administered to all newborn babies in the first seven days after delivery [10]. 

Regardless of BCG vaccination status and local TB incidences, the transmission of TB to the HCWs in health care settings has been reported vastly. Accordingly, the WHO recommends setting a protocol for the prevention of transmitting TB infection to HCWs [11].

The WHO end TB strategy has been taking action to reduce TB incidence by 90% and TB death by 95% until 2035. To achieve this ambitious target in the TB control program, it is paramount to reduce LTBI reservoirs by recommending anti-TB preventive therapy [12]. The WHO recommends three chemoprophylaxis treatment options for LTBI, namely: six months of isoniazid (INH), three months of INH plus rifampicin (RMP), or a three-month regimen of weekly rifapentine plus INH [13].

Iraq is one of the seven highest TB burden countries in the WHO–Eastern Mediterranean Region with a TB incidence of 42 per 100,000 people, constituting 3% of the total TB patients [14].

As there is a high incidence rate of TB in Iraq without the availability of routine LTBI screening for HCWs, the risk of TB reactivation and transmission is a potential threat. Therefore, this study was deemed to be essential to determine the prevalence and associated risk factors of LTBI among HCWs in Duhok province, as well as to prophylactically treat such a vulnerable group from developing TB disease. 

## 2. Materials and Methods

### 2.1. Study Design and Data Recruitment

A cross-sectional prospective study conducted during April–July 2018. HCWs at multi-levels including doctors, pharmacists, dentists, nurses, medical assistants, laboratory technicians, administrative staff, and cleaners were included. HCWs were selected by a non-systematic random sampling method and information on demographic and associated determinant of LTBI were collected by using a standardized questionnaire. Written informed consent was obtained from all staff who agreed to participate in the study. The study was approved by the ethics committee of the Directorate General of Health (DGOH), Duhok, Iraqi Kurdistan (Reference no. 29052018-4, dated 29 March 2018). 

### 2.2. Sample Collection

One mL of blood was drawn by venipuncture into each of the four tubes according to the manufacturer’s instruction, Thereafter, all blood samples from the health care facilities were delivered to the Duhok Central laboratory for further processing.

### 2.3. Laboratory Investigations

The QuantiFERON-TB Gold Plus assay was performed according to the Qiagen manufacturer’s recommendation. Briefly, the QuantiFERON TB Gold plus (QFT-Plus) (Qiagen, Hilden, Germany) assay was incubated at 37 °C overnight. Thereafter, the tubes were centrifuged at 2500 rpm for 15 min. The results were analyzed and calculated based on QFT-GIT analysis software (Version: 2.62). A sample was considered positive if the value of the TB minus Nil control was ≥ 0.35 IU/mL and ≥ 25% of Nil value.

HCWs with indeterminate QFT-Plus results underwent a Tuberculin Skin Test (TST) as per guidelines [15]. The TST was performed by intra-dermal injection of purified protein derivative (PPD) according to Mantoux method [16]. A TST skin reaction ≥ 10 mm was considered positive. All positive HCWs with positive sputum production underwent smear microscopy for acid fast bacilli.

### 2.4. Radiological Examination

All HCWs with positive QFT or positive TST underwent a chest X-ray (CXR) to exclude active TB.

### 2.5. Risk Stratifications

HCWs were stratified according to the following locations for acquiring TB infection [17]
Low risk: administration units, fertility clinic, surgery unit, and the primary health care centers.Moderate risk: pharmacy units and radiography units.High risk: TB healthcare facilities, medical wards, laboratory units, and emergency units.

### 2.6. Statistical Analysis

Data were entered into SPSS software version 22 (Chicago, IL, USA). Categorical variables were presented as frequencies and percentages. Chi square (X^2^) test and fisher exact test were used to find the association between the categorical variables. *p* ≤ 0.05 was considered statistically significant. Odds ratio and binary logistic regression analysis were used to calculate crude and adjusted risk.

## 3. Results

A total of 400 HCWs participated in this study; however, five (1.25%) cases were excluded because they had indeterminate QFT-Plus results and they refused to undergo further diagnostic tests to interpret the results. The study therefore includes 395 (98.75%) HCWs (Figure 1). The mean age of the HCWs was 33.4 ± 9.25 with a female predominance (51.1%). The characteristics of the enrolled staff are shown in Table 1. Overall, 49 (12%) HCWs out of 395 HCWs showed positive tests for LTBI, thereafter referred as IGRA positive. 

As demonstrated in Table 2, according to the univariate logistic regression analysis, when the risk factors associated with positive IGRA were compared to IGRA negative HCWs, significant association was found with the following variables: age group ≥30 years, alcohol intake, ≥11 years of employment, high risk stratification workplaces, and medical doctors.

In the multivariate analysis, the age group of 30–39 years (OR = 0.288, 95% CI: 0.105–0.794, *p* = 0.016) was the only risk factor associated with LTBI. No significant risk factors were identified among other age groups, genders, alcohol intakes, years of employment, risk stratification of workplaces, and medical doctors (Table 3). 

All HCWs with positive LTBI underwent CXR examination, and the results were normal CXR. Those HCWs with positive LTBI, who had sputum, underwent AFB smear microscopy and, subsequently, their results were negative. In conclusion, there were no active TB cases among HCWs with LTBI.

With regards to prophylactic treatment of IGRA positive HCWs (No. = 49), they were offered preventive therapy. Thirty-one (63.3%) accepted the treatment, whereas 18 (36.7%) declined the chemoprophylaxis. Of these 31 HCWs on chemoprophylaxis, 12 (38.7%) received INH for 6 months, and 17 (54.8%) received INH in combination with RMP for three months. To sum it up, 29 (93.5%) HCWs tolerated the treatment without side effects. Whereas, the other two (6.5%) HCWs received alternative therapy because of anti-TB drug intolerance. The drug intolerance was as follows: The first HCW, who had non-alcoholic fatty liver disease (NAFLD), could not tolerate INH because the liver enzymes’ elevation was more than five times the upper limit of the normal; the second HCW did not tolerate INH as jaundice and elevation of liver enzymes were developed. 

## 4. Discussion

Screening and exploring determinants of LTBI among HCWs is an important measure to reduce TB incidence. 

In this study, the prevalence of LTBI in HCWs was 12%, which is lower than the prevalence (27.8%) reported by Al-Lami et al. from 212 HCWs in Baghdad [18]. The discrepancy in the findings between the two studies from different geographic locations in the same country can be explained by that in the latter study, where they screened HCWs at high-risk workplaces and they used TST, which may yield false positive results due to cross-reactivity with BCG-vaccinated people and with those infected with MOTT infection. Additionally, evidence from Duhok National TB Program (NTP) center showed lower TB incidences in comparison to Baghdad, which is also contributed to the low prevalence of TB infection in HCWs.

Considering studies from neighboring countries, the rate was lower compared to Turkey, where 58.8% of HCWs were positive by TST, and 20.6% were positive by IGRA [19]. In addition to Turkey, a study from Iran showed a prevalence of LTBI among HCWs of 16% and 17% using TST and IGRA, respectively [20]. Studies from Saudi Arabia using TST showed different prevalence rates ranging from 11–22.5% in comparison to our study [21,22].

The prevalence of LTBI in Eastern Asian countries varies from country to country. For example, the prevalence of LTBI in this study was higher than that reported from Malaysia [23], Japan [24], which showed rates of 10.6%, and 9.9%, respectively, whereas it was lower than that reported from Taiwan [25], South Korea [26], India [27], and China [28], which showed 14.5%, 17.2%, 31%, and 33.6%, respectively. 

On the whole, the relatively low prevalence rate of LTBI in our study is probably related to the low occupational exposure to active TB cases. In addition, about 50% of the HCWs included in this study have a low mean age of 33.4 ± 9.25 with a duration of less than five years of employment. In the healthcare sector, the prevalence of LTBI increases with the length of employment (>20 years) and also with increasing age (>55 years) [29].

In this study, univariate logistic regression analysis revealed five relevant risk factors of IGRA to be positive. Age groups (30–39, 40–49, and ≥50 years), alcohol intakes, ≥11 years of employment, high risk stratified workplaces, and medical doctors were independent risk factors of LTBI. To exclude possible cofounders in the current study, all variables analyzed in the univariate method were studied in the multivariate model. Correspondingly, one variable, i.e., the age group of 30–39 years (OR = 0.288, 95% CI: 0.105–0.794, *p* = 0.016), was the only significant predictor of LTBI among HCWs. 

The finding of the younger ages in this study was in contrast to most published literature where they stated that the risk of LTBI is directly linked with increasing age [30,31]. They documented the reason behind age increasing with LTBI as relating to a prolonged exposure to *Mtb* infection. The precise explanation of why LTBI was high in 30–39 years is vague in this study, but it may be due to including higher numbers of this age group. As a consequence, further research is warranted to better understand the association of LTBI with age.

The finding of alcohol consumption was not significant in this study by the multivariate method. However, it requires meticulous attention as there may be an underestimation of its report as the community in Duhok is socially and religiously conservative about alcohol intake. A systematic review and meta-analysis by Simou et al. [32] about alcohol consumption and the risk of TB infection concluded that alcohol is a substantial risk factor for developing TB. As a matter of fact, alcohol consumption contributes to impaired immunity, hence, increasing the risk of acquiring TB infection and, accordingly, reactivating LTBI [33].

In the current study, although ≥11 years of employment and high risk stratified workplaces were not independent risk factors by multivariate analysis, they were significantly associated with LTBI in the univariate analysis. These risk factors are important predictors of LTBI among HCWs, which were consistent with the findings of other studies [26,28]. It is well known that more years of employment and working in high-risk TB places are important cumulative factors to acquire TB infection [17]. Hence, with our limited health resources, it is recommended to improve infection control measures at least at high-risk workplaces. 

In this study, with regards to occupation, although the medical doctors’ group was not an independent risk factor with LTBI, it was significant by the univariate model. This finding was partially comparable to findings of a Turkish study, which reported a significant association between doctors and LTBI [34]. Another study commenced in Northern Germany by Schablon and colleagues [35], in a hospital of pulmonary diseases, showed a higher prevalence of LTBI among doctors and nurses. In this study, although the prevalence of LTBI among nurses was high (32.6%), it was not statistically significant by univariate model. This can be explained by pointing out that several nurses in our locality are non-university graduates and they perform administrative jobs rather than clinical jobs. On the other hand, doctors work in out-patient and in-patient healthcare facilities, which cause them to be more prone to infections, including TB. 

In our study, we did not find any case of active TB among HCWs with LTBI. This finding was in concordance to a study from Iran [20]. Apparently, the high socioeconomic status of the HCWs, in comparison to the general population has been the cause of low TB incidences in the health staff. 

In the present study, unpredicted high rate (36.7%) HCWs who declined anti-TB chemoprophylaxis was found. This can be partially explained as symptoms and signs of TB are lacking. Despite that, HCWs who lack sufficient medical knowledge refuse medicine intake by claiming that INH harms the liver. The refusal rate in the current study was exceedingly high in comparison to some studies [36], whereas it was low in comparison to others [37]. Notably, acceptance of anti-TB chemoprophylaxis is low among health employees in comparison to the general population [37,38]. Therefore, ensuring medical efforts to educate the healthcare staff particularly non-professionals, is a priority to encourage chemoprophylaxis acceptance. In the present study, the majority (54.8%) of employees preferred a three month, short-course chemoprophylaxis INH/RMP prescription, whereas 36.7% received INH alone for six months. This is in agreement with other studies where HCWs preferred to receive a short-course anti-TB preventive therapy [37]. As a result, short course chemoprophylaxis of INH/RMP should be offered for HCWs as it is both time and treatment effective compared to the standard 6–9 month regimen INH. 

The response to therapy was convenient, as the majority of HCWs (93.5%) tolerated the treatment. On the other hand, 6.5% employees did not tolerate the treatment because of hepatoxicity that warranted anti-TB drug withdrawal, and hence, an alternative regimen was used for the prophylaxis. Our finding was nearly compatible with the findings of other studies who used INH/RMP for chemoprophylaxis [39]. In this study, we did not recommend the use of RMP monotherapy, as Iraq is a relatively high TB burden country and we would like to keep RMP in reserve for active TB cases, henceforth, to avoid RMP resistance and subsequent emergence of multi-drug resistant-TB.

There were few limitations in this study. First, the sample size was not large enough to represent all HCWs in the whole province. Second, the classification of risk stratification may have had potential bias in this study as several HCWs have a history of working in other departments prior to their current workplaces. Third, the risk presentation of alcohol intake was not precise because of the socio-religious barrier. Fourth, there were some biases, such as refusal of chemoprophylaxis, hence, we were not able to accurately emphasize the results. Furthermore, the cross-sectional nature of the study is not without limitation with associated variables. Therefore, larger sample size studies with a control group are warranted to better understand such association. 

In conclusion, although Iraq is a relatively high TB burden country, the prevalence of LTBI among Duhok HCWs is low to some extent. It is important to screen HCWs in Duhok for LTBI particularly medical doctors, young adults, alcoholics, and those whom had a long duration of employment in high risk workplaces. The acceptance rate of HCWs with LTBI to chemoprophylaxis was low. Therefore, ensuring medical efforts to educate the healthcare staff, particularly non-professionals, is a priority to encourage chemoprophylaxis acceptance.

## Figures and Tables

**Figure 1 tropicalmed-04-00085-f001:**
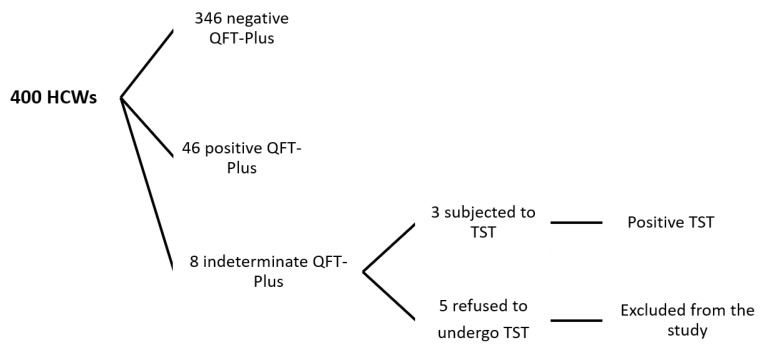
Flowchart of the studied population.

**Table 1 tropicalmed-04-00085-t001:** Characteristics of the healthcare workers.

Parameter	No. (%) *
**Age**	
Mean ± SD	33.4 ± 9.25
**Gender**	
Male	193 (48.9)
Female	202 (51.1)
**Cigarette smokers**	
Yes	75 (19.0)
No	320 (81.0)
**Alcohol consumption**	
Yes	18 (4.6)
No	377 (95.4)
**Diabetics**	
Yes	15 (3.8)
No	380 (96.2)
**BCG vaccination**	
Yes	384 (97.2)
No	11 (2.8)
**History of TB contact**	
Yes	38 (9.6)
No	357 (90.4)
**Years of employment**	
Mean ± SD	9.09 ± 8.39
**Nationality**	
Iraqi	387 (98.0)
Syrian	6 (1.5)
Turkish	2 (0.5)
**Occupation**	
Medical doctors	38 (9.6)
Pharmacists	11 (2.8)
Dentists	4 (1.0)
Nurses	132 (33.4)
Paramedics	92 (23.3)
Laboratory Technicians	60 (15.2)
Administrative staff	39 (9.9)
Cleaners	19 (4.8)

* Mean ± SD was used for age and length of employment.

**Table 2 tropicalmed-04-00085-t002:** Univariate analysis of associated risk factors with positive IGRA results.

Variable	IGRA Positive (No. = 49) No. (%)	IGRA Negative (No. = 346) No. (%)	Total (No. = 39) No. (%)	Odd Ratio (95% CI)	*p* Value
**Age group (year)**					
≤29 *	10 (20.4)	163 (47.1)	173 (43.8)	-	-
30–39	21 (42.9)	99 (28.6)	120 (30.4)	3.46 (1.56–7.64)	0.0013
40–49	13 (26.5)	63 (18.2)	76 (19.2)	3.36 (1.40–8.06)	0.0045
≥50	5 (10.2)	21 (6.1)	26 (6.6)	3.88 (1.21–12.45)	0.0154
**Gender**					
Male	30 (61.2)	163 (47.1)	193 (48.9)	1.77 (0.96–3.27)	0.06
Female	19 (38.8)	183 (52.9)	202 (51.1)		
**Cigarette smokers**					
Yes	9 (18.4)	66 (19.1)	75 (19)	0.96 (0.44–2.06)	0.91
No	40 (81.6)	280 (80.9)	320 (81)		
**Alcohol consumption**					
Yes	5 (10.2)	13 (3.8)	18 (4.6)	2.91 (0.99–8.56)	0.05
No	44 (89.8)	333 (96.2)	377 (95.4)		
**Diabetics**					
Yes	2 (4.1)	13 (3.8)	15 (3.8)	1.09 (0.24–4.98)	0.91
No	47 (95.9)	333 (96.2)	380 (96.2)		
**BCG status**					
Yes	48 (98)	336 (97.1)	384 (97.2)	1.43 (0.18–11.41)	0.74
no	1(2)	10 (2.9)	11 (2.8)		
**History of close TB contact**					
Yes	6 (12.2)	32 (9.2)	38 (9.6)	1.37 (0.54–3.47)	0.51
No	43 (87.8)	314 (90.8)	357 (90.4)		
**Nationality**					
Iraqi	48 (98)	339 (98)	387 (98)	NA	0.25 **
Syrian	0 (0)	6(1.7)	6 (1.5)		
Turkey	1 (2)	1 (0.3)	2 (0.5)		
**Years of employment**					
≤1 *	2 (4.1)	45 (13)	47 (11.9)	–	–
2–5	13 (26.5)	133 (38.4)	146 (37)	2.20 (0.48–10.12)	0.301
6–10	11 (22.5)	67 (19.4)	78 (19.7)	3.69 (0.78–17.46)	0.099
≥11	23 (46.9)	101 (29.2)	124 (31.4)	5.12 (1.16–22.67)	0.018
**Risk stratification of workplaces**					
Low *	8 (16.3)	95 (27.4)	103 (26.1)	-	-
Moderate	3 (6.1)	53 (15.3)	56 (14.2)	0.67 (0.17–2.642)	0.57
High	38 (77.6)	198 (57.2)	236 (59.7)	2.28 (1.02–5.07)	0.044
**Occupation**					
Medical doctors	9 (18.4)	29 (8.4)	38 (9.6)	2.46 (1.09–5.57)	0.03
Pharmacists	2 (4.1)	9 (2.6)	11 (2.8)	1.59 (0.33–7.60)	0.55
Dentists	0 (0)	4 (1.2)	4 (1)	0.77 (0.05–14.50)	0.86 **
Nurses	16 (32.6)	116 (33.5)	132 (33.4)	0.96 (0.51–1.82)	0.9
Paramedics	8 (16.3)	84 (24.3)	92 (23.3)	0.61 (0.27–1.35)	0.22
Laboratory technicians	6 (12.3)	54 (15.6)	60 (15.2)	0.76 (0.31–1.86)	0.54
Administrative staff	3 (6.1)	36 (10.4)	39 (9.9)	2.70 (0.93–7.84)	0.59
Cleaners	5 (10.2)	14 (4)	19 (4.8)	0.56 (0.17–1.90)	0.35

* Reference group. ** Fisher exact test.

**Table 3 tropicalmed-04-00085-t003:** Multivariate analysis of associated risk factors with positive IGRA result.

Variables	Odd Ratio (95% CI)	*p* Value
**Age group**		
≤29	-	0.121
30–39	0.288 (0.105–0.794)	0.016
40–49	0.330 (0.100–1.090)	0.069
≥50	0.332 (0.074–1.499)	0.152
**Alcohol consumption**		
Yes	0.325 (0.098–1.077)	0.066
No		
**Employment years**		
≤1	-	0.920
2–5	0.590 (0.120–2.889)	0.515
6–10	0.545 (0.094–3.164)	0.499
≥11	0.546 (0.091–3.256)	0.506
**Risk stratification of workplaces**		
Low	-	0.150
Moderate	0.563 (0.177–1.792)	0.331
High	0.436 (0.188–1.007)	0.052
**Job as a doctor**	0.468 (0.203–1.082)	0.076

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
