# Peer review of "Latent Tuberculosis Infection among Healthcare Workers in Duhok Province: From Screening to Prophylactic Treatment"

_tropicalmed, 2019, doi:10.3390/tropicalmed4020085_

Round 1

Reviewer 1 Report

General comment:

It is well known that one-third of the world is infected with latent Mtb. 10% cases could possibly get re-activation and cause active TB. The manuscript reported latent TB cases amongst the healthcare worker (HCW) in Duhok, Iraq. The authors found that the prevalence of Latent TB (LTBI) was at low12%. Using Quantiferon-TB Gold plus assay, they identified 49 HCWs out of 395 participant. Using Tuberculin Skin test, they confirmed that three subjects were TB positive while five refused to be tested. Overall, the study is an important report to alert and manage a better healthcare system in the area under investigation. But, there were many concerns with the author's claim, conclusion and references. Scientific writing and Proof reading was poor throughout the manuscript.

Specific Comments:

References 3,8,13,14,15,16,17,18,19,20,21,22,23,24,25,30,31,33 had the similar kind of study at different geographical region in the world. It clearly implies that either there is improper screening of HCW or there is lack of regulations/compliance. I have specific questions to the authors as follows:

Line 2: The entire manuscript discuss and report about LTB, then why did the author uses TB in the topic? 

Line 21: Why didn't the authors just mention age group above 30 years instead?

Line 34,35: Are these non professionals included in any of the TB hospitals? 

Line 99: Why does the authors consider lab units as moderate? That is the main sample collection and processing?

Line 126, Table 2: Interesting that 48 IGRA+ HCW were BCG vaccinated. Any thoughts by the authors? How many times was the IGRA assay performed? How many false positive reported?

Line 148: It is not true if the authors want to claim the study to be the first in Iraq, since they just used an advance IGRA assay compared to previous report by Al-Lami et al.

Line 151: I don't agree with the comparison in prevalence since Al-Lami et al, had specifically screen for three TB healthcare, where there was high chance of TB exposure. Why did the authors think that it was the similar case?

Line 154: If TST is imprecise and non-specific then why did the authors use in the study? 

Line 156: What is Duhok NTP?

Line 206-208: What does the authors want to convey in this statement? TB is the number one killer pathogen, does the education of the nurse not a high priority to handle and contain the possible exposure? 

Line 248: Does the healthcare system/ hospitals have an annual health check up on each HCW? Who are responsible for the monitoring and compliance?

Minor corrections:

Line 9: Upper case, Mycobacteria

Line 10: Italics Mtb....

Line 38: TB was declared ....

Line 46: Delete one full stop..

Line 83: Mention Qiagen here instead of putting in the reference.

Author Response

Response-Reviewer 1:

Thank you for the kind comments and advices

Answer to specific comments:

Line 2: I have changed the topic from tuberculosis infection to latent tuberculosis infection, hence the title changed to "Latent Tuberculosis Infection among Healthcare Workers in Duhok Province: From Screening to Prophylactic Treatment"

Line 21: I have changed the age group of 30-39, 40-49, and ≥ 50 years into age groups of ≥ 30 years. I made the same changes in the results with regards to line 122.

Line 34, 35: Yes, there was only one non-professional included in a TB center.

Line 99: Thank you for the comment. We reviewed other papers [1], where they also included Lab. units in high risk workplaces. Therefore, we followed the reviewer's instructions and we had made the changes accordingly as shown in the followings: stratification in the materials and methods (now laboratory units are included with high risk workplaces), and tables 2 and 3. The statistical analysis has been changed in the manuscript and everything is highlighted clearly in the paper.

Line 126: We performed IGRA test once for BCG vaccinated HCWs, as it is known positive IGRA does not cross react with BCG vaccination.

Line 148: Yes, we totally agree that this was not the first study about LTBI in Iraq but it is thought to be the first study using advanced IGRA assay. Anyway, we have deleted the sentence of "This is the first insight study about HCWs related to LTBI in Iraq using new generation IGRA." from the manuscript. 

Line 151: We don't think it was a similar case, but we compared our study with Al-Lami since it is from Iraq (same country). Furthermore, we mentioned that the prevalence was higher in their study because they screened HCWs at high risked workplaces, and they used TST screening which might cross react with Mycobacterium other than TB (MOTT) infection and BCG vaccination, and hence the prevalence of LTBI was higher in their study.

Line 154: We used the term "imprecise" inaccurately. We meant that TST was not specific as it may show false positive results due to cross reactivity with BCG vaccination and also with nontuberculous mycobacterial infection. We have deleted the term "imprecise" and we have changed that in the discussion and it is highlighted in the manuscript.

Line 156:  NTP stands for National TB Program (NTP) Center in Duhok

Line 206-208: Of course health education of the nurses is a top priority. We are in the infection control team working hardly to educate the nurses about the importance of TB. However, we meant that the health system in Iraq including Duhok is quite different from the majority of the rest of the world because most of the nurses were non university graduates. They are technical nurses [2] without sufficient medical knowledge. The College of Nursing in Duhok has been established during 2007-2008 with a limited number of graduates, which does not meet the demand of Duhok province.   

Line 248: We have Infection Control Teams (ICT) in tertiary hospitals e.g. Azadi Teaching Hospital, but we do not have regular annual health checks for each HCW because this is not a part of routine regulations in our health system. 

Minor corrections:

Line 9: mycobacterium tuberculosis has been changed into upper case "Mycobacterium tuberculosis"

Line 10: Mtb was made italic "Mtb". We changed all Mtb terms to italic style throughout the manuscript.

Line 38: TB is declared has been changed to TB was declared.

Line 46: additional full stop was deleted.

Line 83: "Qiagen" was added instead of putting in the reference. 

References:

[1] Joshi, R., et al., Tuberculosis among health-care workers in low-and middle-income countries: a 280 systematic review. PLoS medicine, 2006. 3(12): p. e494.

[2] Garfield, R., Dresden, E., & Boyle, J. S. (2003). Health care in Iraq. Nursing outlook, 51(4), 171-177.

Reviewer 2 Report

Minor changes: Table 1 needs editing-nationality appears twice

One question: If as stated in the text, “Iraq is a relatively high TB burden country”, why were the Healthcare workers not chosen from Institutions where high TB exposure occurs. They would be at higher risk of developing latent TB Infection. According to the data presented, only about 9% had history of TB contact.

Author Response

Response-Reviewer 2:

Thanks for your kind comments and advices

Minor changes:

1-      We deleted the extra nationality in the table as it was appearing twice.

2-   We have chosen HCWs from different institutions at multi-level risk of exposure to get a representative prevalence for all HCWs. We avoided considering high risk HCWs as to avoid representing prevalence of high risk workplaces only.

Reviewer 3 Report

Article:  Tuberculosis infection among healthcare workers in Duhok province: from screening to prophylactic treatment.

This is a current topic in the times of tuberculosis (TB) as an increasing global emergency. Healthcare workers (HCWs) are at a very high risk of latent TB infection (LTBI) and TB disease, because of their sustained occupational exposure.   

I have read the paper carefully, and found that it could be acceptable for publishing after mayor and minor essential revisions. All the comments are detailed here in this document.

Major Observations:

1.     Lines 23-25; 129-131; 178-180: “In the multivariate analysis, age group of 30-39 (OR = 0.347, 95% CI: 0.130-0.932, P value = 0.036), and medical doctors (OR = 0.437, 95% CI: 0.190-1.003, P value = 0.05) were the only risk factors associated with LTBI”. This conclusion is inaccurate. A p value ≤0.05 is a standard cut-off point for claiming results to be statistically significant, this means that there is ≤5% probability of an effect to happen by chance. Although, according to the study results (p = 0.05) we would have to reject the null hypothesis, which means that the outcome found in the exposed group (risk of having LTBI) does differ from that in the control group. But, 95% CI (0.190-1.003) is too wide and includes the null value of 1 (Figure 1). Then, there is insufficient evidence to conclude that the groups are statistically significantly different.  Moreover, statistically significant does not mean clinically significant.

2.     As the authors state in Lines 43-45 there are well known risk factors, mainly HIV coinfection, which could potentiate the reactivation of LTBI. So, it would be appropriate to know Healthcare workers (HCWs) HIV status. HIV status could have profound and definite implications. The risk of progression from LTBI to TB disease is 7-10% each year for those with both LTBI and untreated HIV infection. Those with LTBI who are not HIV infected have a 10% risk over their lifetime. If HIV status could not be tested, for example because of religious reasons, it must be clearly explained in Limitation of the study (1).  

3.     Lines 60, 61: “Iraq is one of the 7 highest TB burden countries in the WHO-Eastern Mediterranean Region with 60 a TB incidence of 42 per 100,000 people, constituting 3% of the total TB patients”. The authors explain that Iraq is a country of high endemicity of TB. Besides, even though the authors do not specify if Iraqi population is routinely vaccinated with Bacillus Calmette–Guérin (BCG), this appear to be the case, because in Table 1 and Table 2 we can see that 97.2% of participants were BCG vaccinated. This fact should be clearly specified by the authors.

4.     The authors do not explain why they chose to perform IGRA instead of TST as the initial test considering that Iraq is a low-income country. Although, there is no gold standard test for diagnosis of LTBI, and both Tuberculin skin test (TST) and blood interferon-gamma release assay (IGRA) tests can be performed to diagnose LTBI (2-4). Although, what the authors cite in Lines 154-156 (“TST is not specific, and it may have showed false positive results in BCG vaccinated people, and other mycobacterial infections might have been mistaken for M. tuberculosis”) is true, the CDC states that considering comparable performance between TST and IGRA but increased cost, replacing TST with IGRAs in low-income and other middle-income countries is not recommended (2). The WHO recommends that Bacillus Calmette–Guérin (BCG) vaccine be administered during infancy in TB endemic countries. In addition, the Center for disease control (CDC) guidelines state that TST reactivity caused by BCG vaccine generally wanes with the passage of time and a person with a history of BCG vaccination can be tested and treated for LTBI if they react to the TST. TST reactions should be interpreted based on risk stratification regardless of BCG vaccination history (2).

5.     The authors do not explain why to perform TST when IGRA was “indeterminate”. As a matter of fact, 8 patients with “indeterminate” QFT-plus test were subsequently tested with TST. According to the authors IGRA gives “an improved outcome in detecting specific infection of Mtb complex (Lines 52, 53). Indeed, IGRA is generally considered to be more specific but with similar sensitivity compared with TST (2-4). Furthermore, British guidelines recommend using IGRA tests as confirmatory tests in those with positive TST results (4). In addition, a cost–analysis study in Germany found that screening of contacts by TST followed by QFT assay in positive TST reactors was the most cost-effective method for screening for latent TB. In countries where the BCG vaccination rate is high, this two-step approach is probably more efficient for screening of high-risk individuals (e.g., health care workers) because, it improves the diagnostic accuracy and reduces false positive results (5) and this is likely to have a significant positive impact on the control of latent TB (2, 3).

6.     Lines 143, 144: The authors say that 2 patients (6.5%) received alternative therapy because of anti-TB drug intolerance. What type of intolerance? The authors should explain this in detail.

7.     Lines 228: the authors state “The response to therapy was convenient”. However, they do not provide information in what sense this response to therapy was convenient. In fact, in Results the authors do not show any result related with response to therapy. They just say in Line 144 that 6.5% of participants received alternative therapy because of anti-TB drug intolerance, and in Lines 228 and 229 they added that these employees did not tolerate the therapy because of hepatoxicity. The authors do not specify if these “hepatotoxicity” or “drug intolerance” reactions warranted anti-tuberculous drug withdrawal.

Minor Observations

1.     Line 215, 216: “…symptoms and signs of TB are lacked”. Since, this description refers to LTBI, a better definition would be: “…symptoms and signs of TB are lacking or absent”. Please correct misspelling mistake.

2.     I would please you to revise the reference list according to this journal style (https://www.mdpi.com/journal/tropicalmed):

Pappas, G.; Papadimitriou, P.; Akritidis, N.; Christou, L.; Tsianos, E.V. The new global map of human brucellosis. Lancet. Infect. Dis. 20066, 91–99.

NOT:

Steele, M.A., R.F. Burk, and R.M. DesPrez, Toxic hepatitis with isoniazid and rifampin: a 339 meta-analysis. Chest, 1991. 99(2): p. 465-471.

References:

1.        Janagond AB, Ganesan V, Vijay Kumar GS, Ramesh A, Anand P, Mariappan M. Screening of health-care workers for latent tuberculosis infection in a Tertiary Care Hospital. Int J Mycobacteriol. 2017; 6(3):253-257. doi: 10.4103/ijmy.ijmy_82_17.

2.        Centre for Disease Control and Prevention. Latent Tuberculosis Infection: A Guide for Primary Health Care Providers. Atlanta, Georgia: CDC; 2014.

3.        Al-Orainey IO. Diagnosis of latent tuberculosis: Can we do better? Ann Thorac Med. 2009;4(1):5–9. doi:10.4103/1817-1737.44778

4.        National Collaborating Centre for Chronic Conditions. Tuberculosis: Clinical diagnosis and management of tuberculosis, and measures for its prevention and control. London: Royal College of Physicians; 2006

5.        Diel R, Loddenkemper R, Meywald-Walter K, Niemann S, Nienhaus A. Predictive value of a whole-blood IFN-y assay for the development of active tuberculosis disease after recent infection with Mycobacterium tuberculosis. Am J Respir Crit Care Med. 2008;177:1164–70. doi: 10.1164/rccm.200711-1613OC.

Author Response

Response-Reviewer 3:

Thank you for your kind comments and advices.

Response to major observations:

1.      With regards to line 23-25; 129-131; 178-180: We totally agree with the multivariate analysis and inaccuracy of the significance of medical doctors. In the revised version of the manuscript, the first reviewer suggested to include laboratory workers with the high risk workplaces, hence the statistical analysis has been changed markedly in the manuscript. In the revised version, according to the multivariate analysis medical doctors were not associated risk factor with LTBI. The changes are highlighted in the paper.

2.      Lines 43-45, there is very low prevalence HIV status among HCWs. We have added that to the introduction " Local studies from Iraqi Kurdistan showed very low prevalence of HIV infection among both general population and specific populations such as HCWs [1, 2]"

3.      Lines 60, 61: yes there is a routine universal vaccination of all new born babies during the first week of birth. We have added that to the introduction "In Iraq, as part of national immunization program, BCG vaccine is routinely administered to all new born babies in the first 7 days after delivery [3]."

4.      We used IGRA in this study, as it was a part of MSc project. The scientific committee accepted to perform the study for the student by introducing new techniques for diagnosing LTBI among HCWs. TST is not available regularly in our TB center of Duhok and on many occasions; there is a deficiency of PPD in the center. Additionally, we wanted to avoid false positive results due to MOTTs and BCG vaccination, hence we decided to use IGRA.

5.      Indeed, we used TST for indeterminate IGRA results, according to recommendations by CDC fact sheet (https://www.cdc.gov/tb/publications/factsheets/testing/igra.htm) and also according to the article by Castro et at. (2010) [4]. We did not repeat IGRA tests because it was costly. We have added reference [4] for that in the materials and methods.

6.      With regards to type of drug intolerance, the HCWs recommended short course combination therapy of INH plus RMP as it is not time consuming. We advised this regimen for young HCWs. We did not recommend combination therapy for elder HCWs and we advised them to receive INH chemoprophylaxis to avoid side effects. In the results, we have added " The drug intolerance was as follows: The first HCW, who had non alcoholic fatty liver disease (NAFLD), could not tolerate INH because the liver enzymes' elevation was more than 5 times the upper limit of the normal; the second HCW did not tolerate INH as jaundice and elevation of liver enzymes were developed."

7.      The response to therapy was convenient as the majority of HCWs (93.5%) tolerated the treatment without side effects. We added this clarification in the results "To sum it up, 29 (93.5%) HCWs tolerated the treatment without side effects." We have clearly specified that anti-drug side effect (hepatotoxicity) warranted discontinuation of the drug and then starting an alternative therapy. We added that in the discussion "On the other hand 6.5% employees did not tolerate the treatment because of hepatoxicity that warranted anti-TB drug withdrawal, and hence, an alternative regimen was used for the prophylaxis"

Response to  minor Observations:

1.      Line 21, 216: we changed …symptoms and signs of TB are lacked to "…symptoms and signs of TB are lacking"

2.      We revised the references according to the journal's style based on endnotes.

References:

[1] MA Merza et al. Low prevalence of hepatitis B and C among tuberculosis patients in Duhok Province, Kurdistan: Are HBsAg and anti-HCV prerequisite screening parameters in tuberculosis control program?. Int. J. Mycobacteriol. (2016

[2] Hussein, Nawfal R. "Prevalence of HBV, HCV and HIV and Anti-HBs antibodies positivity in healthcare workers in departments of surgery in Duhok City, Kurdistan Region, Iraq." International Journal of Pure and Applied Sciences and Technology 26.2 (2015): 70.

[3] World Health Organization. Guidance for national tuberculosis programmes on the management of tuberculosis in children. No. WHO/HTM/TB/2014.03. World Health Organization, 2014.

[4] Castro, Kenneth G., et al. "Updated guidelines for using interferon gamma release assays to detect Mycobacterium tuberculosis infection--United States, 2010." (2010).

Round 2

Reviewer 1 Report

The revised version looks better now and I would agree with your response regarding my concern. Hope that a better regulatory system will be alerted based on this report. Good luck!

Author Response

Thank you for the kind comments and advices. I am happy to find you considering the new version better than the last version.

Reviewer 3 Report

The authors have addressed my concerns and provided enough detail in explanations in the revised version of the manuscript. I’m totally satisfied with the new version and have no additional observations.

Author Response

Thank you for your kind comment. I am glad that you are satisfied with the new version.
